# Experimental witness of quantum jump induced high-order Liouvillian exceptional points

Zhuo-Zhu Wu[1,2,10], Pei-Dong Li [1,2,10], Tai-Hao Cui[1,2], Jia-Wei Wang[1,2], Yuan-Zhang Dong[1,2], Shuang-Qing Dai [1,2], Ji Li [3], Ya-Qi Wei[4], Quan Yuan[5], Xiao-Ming Cai [1], Liang Chen [1] ✉, Jian-Qi Zhang [1] ✉, Hui Jing [6,7,8] ✉ & Mang Feng [1,3,6,9] ✉

The exceptional point has presented considerably interesting and counterintuitive phenomena associated with nonreciprocity, precision measurement, and topological dynamics. The Liouvillian exceptional point (LEP), involving the interplay of energy loss and decoherence inherently relevant to quantum jumps, has recently drawn much attention due to capability to fully capture quantum system dynamics and naturally facilitate non-Hermitian quantum investigations. It was also predicted that quantum jumps could give rise to third-order LEPs in two-level quantum systems for its high dimensional Liouvillian superoperator, which, however, has never been experimentally confirmed until now. Here we report the observation of the third-order LEPs emerging from quantum jumps in an ultracold two-level trapped-ion system. Moreover, by combining decay with dephasing processes, we present the experimental exploration of LEPs involving combinatorial effect of decay and dephasing. In particular, due to non-commutativity between the Lindblad superoperators governing LEPs for decay and dephasing, we witness the movement of LEPs driven by the competition between decay and dephasing occurring in an open quantum system. This unique feature of non-Hermitian quantum systems paves a new avenue for modifying nonreciprocity, enhancing precision measurement, and manipulating topological dynamics by tuning the LEPs.

Degeneracy appears ubiquitously in quantum systems, giving rise to rich physical phenomena. In non-Hermitian quantum systems, spectral degeneracies manifest as the coalescence of two or more eigenvalues along with their associated eigenvectors, known as exceptional points (EPs)[1-3]. There have been many discussions of operations around the EPs, which lead to counterintuitive phenomena[4], such as modifying nonreciprocal transmissions[5-10], and provide potential applications in enhancing precision measurement[11-14]. Moreover, EPs represent the

[1]Wuhan Institute of Physics and Mathematics, Innovation Academy of Precision Measurement Science and Technology, Chinese Academy of Sciences, Wuhan, China. [2]School of Physics, University of the Chinese Academy of Sciences, Beijing, China. [3]Research Center for Quantum Precision Measurement, Guangzhou Institute of Industry Co. Ltd, Guangzhou, China. [4]Laboratory of Quantum Science and Engineering, South China University of Technology, Guangzhou, China. [5]Guangzhou Institute of Industrial Intelligence, Guangzhou, China. [6]Key Laboratory of Low-Dimensional Quantum Structures and Quantum Control of Ministry of Education, Hunan Normal University, Changsha, China. [7]Department of Physics and Synergetic Innovation Center for Quantum Effects and Applications, Hunan Normal University, Changsha, China. [8]Institute for Quantum Science and Technology, College of Science, National University of Defense Technology, Changsha, China. [9]Department of Physics, Zhejiang Normal University, Jinhua, China. [10]These authors contributed equally: Zhuo-Zhu Wu, Pei-Dong Li. ✉e-mail: liangchen@wipm.ac.cn; changjianqi@gmail.com; jinghui73@foxmail.com; mangfeng@wipm.ac.cn

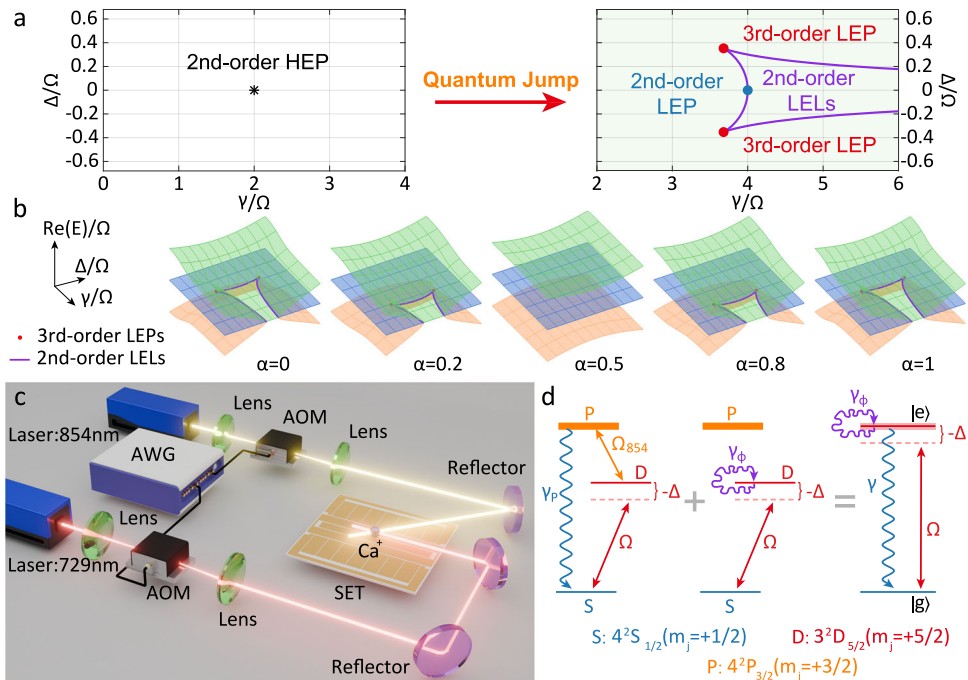

**Fig. 1 | Principle of the experiment for observing the movement of Liouvillian exceptional lines and points. a** Transition from a second-order HEP to third-order LEPs with quantum jump. The second-order HEP (black) splits into two third-order LEPs (red) and three distinct second-order Liouvillian exceptional lines (LELs in purple). **b** Variation of the second-order and third-order LEPs with respect to $\alpha$. The colored surfaces are real parts of the complex eigenenergy Riemann surfaces with the orange, blue and green surfaces corresponding, respectively, to the eigenenergies $E_1$, $E_2$ and $E_3$. Three distinct second-order LELs, depicted in purple, are formed by the second-order LEPs, the third-order LEPs are localized at the crossing points where two of these second-order LELs intersect. **c** Schematic of the experimental setup with a single ultracold $^{40}\mathrm{Ca}^+$ ion confined in a surface electrode trap (SET). AOM: acousto-optic modulator. AWG: arbitrary waveform generator. The AWG generates noise signals that are injected into the 729nm laser beam, which brings in dephasing. Another AOM is employed to control the power of the 854 nm laser, which leads to decay. The 3D schematic was created with Blender software. **d** Level scheme of the ion, where the lefthand side of the equator illustrates the experimental system with the mixture of a pure decay and a pure dephasing and the righthand side is for the equivalent two-level model. The solid lines with double arrows denote the transitions with Rabi frequencies driven by 729 nm and 854 nm lasers, and $\Delta$ is the detuning of the 729 nm laser from the two-level resonance.

points for topological phase transitions on the complex-eigenenergy Riemann surfaces[15–19]. Controlling systems to evolve around or cross the EPs could bring about interesting observations[20–22], such as topological energy transfers[8], asymmetric mode switching[7,23,24] and even thermodynamic effects[25,26], which help us further understand the unique characteristics of non-Hermitian quantum systems[27–29].

Although the intriguing properties of EPs have been extensively explored using effective non-Hermitian Hamiltonians, such Hamiltonian EPs (HEPs) primarily describe coherent dynamics by excluding quantum jump terms[30,31]. To fully capture the dynamics of a quantum system, including the influence of quantum jumps, one should turn to the Liouvillian superoperator and define EPs as degenerate eigenvalues of the Liouvillian superoperators, termed Liouvillian EPs (LEPs)[32–40]. Compared to conventional HEPs, where the highest possible order is constrained by the Hamiltonian's subspace dimensionality[41], LEPs can own the maximum order exceeding that of HEPs due to non-unitary dynamics induced by quantum jumps[42]. For instance, in a two-level or two-mode system, a single second-order HEP can be achieved by its Liouvillian superoperator[32], whereas the third-order LEPs, within the same Liouvillian superoperator subspaces, emerge only in the presence of quantum jumps (see Fig. 1a). Since they exhibit stronger sensitivity and more complex topological characteristic than the second-order HEPs, the third-order HEPs have been experimentally utilized to improve precision measurements[43,44], demonstrate third-order exceptional line[45,46], and explore topological properties[18,47]. However, existence of such high-order LEPs has not yet been experimentally confirmed in quantum systems. There is a strong desire to experimentally demonstrate the third-order LEPs in two-level quantum systems.

The Liouvillian superoperator of the LEPs originates from Lindblad master equation describing dynamics of open quantum systems. For a two-level atom, the Lindblad master equation typically includes two primary dissipative processes: decay and dephasing. Consequently, the LEPs in such a quantum system can be classified into three distinct types: (i) LEPs only arising from decay processes[34,36–39], (ii) LEPs only induced by dephasing processes[40], and (iii) LEPs resulting from combination of both decay and dephasing processes, which can be called Lindblad LEPs. However, while previous experiments focus on the LEPs due to either the decay or dephasing, the Lindblad LEPs, which are fully consistent with the Lindblad master equation for open quantum systems, have never been investigated experimentally.

In this work, we experimentally explore the variation of LEPs, from purely dephasing case to Lindblad LEPs and then to purely decay case, by adjusting both decay and dephasing effects. Our experimental results demonstrate the observation of the third-order LEPs in a two-level quantum system as well as their movement with respect to the ratio of decay and dephasing rates. Both the LEPs due to decay and dephasing share the identical parameters (i.e., $\Omega$ and $\gamma$ as defined later), suggesting that the Lindblad LEPs are also with the same parameters. However, as elucidated later, competition of the decay rate $\gamma_0$ and the dephasing rate $\gamma_\phi$ brings about movement of the LEPs on the Riemann surfaces. The key point is that the Lindblad superoperators for decay and dephasing do not commute with each other, resulting in the movement of LEPs with respect to the ratio of decay and dephasing rates.

By elaborately tuning the proportion of $\gamma_0$ and $\gamma_\phi$, i.e., $\gamma_0/\gamma_\phi = \alpha/(1 - \alpha)$, we demonstrate the existence of the third-order LEPs for dephasing at $\alpha = 0$ ($\gamma_0 = 0$) and for decay at $\alpha = 1$ ($\gamma_\phi = 0$) in an effective

two-level atom. Moreover, different from current studies about LEPs either neglecting the influence from decay or ignoring the effect of dephasing, we observe the third-order Lindblad LEPs under the government of both decay and dephasing effects. Additionally, with the competition between $\gamma_\phi$ and $\gamma_0$, we experimentally witness the variation of the second- and third-order LEPs with respect to $\alpha$. In particular, our study reveals that the Lindblad LEPs disappear at $\alpha = 1/2$ ($\gamma_\phi = \gamma_0$) since they are shifted to infinity in the parameter space.

## Result

### Experimental model and setup

Before presenting the experimental details, we first elucidate our model with a two-level system consisting of an upper state $|e\rangle$ and a lower state $|g\rangle$. The dynamics of such a non-Hermitian quantum system is described by $\dot{\rho} = \mathcal{L}[\rho]$, where $\rho$ denotes the density operator, and $\mathcal{L}$ is a Lindblad superoperator given by ($\hbar = 1$),

$$\mathcal{L}[\rho] = -i[H, \rho] + \gamma_0 \mathcal{L}_{|g\rangle\langle e|}[\rho] + \gamma_\phi \mathcal{L}_{|e\rangle\langle e|}[\rho]. \quad (1)$$

Here $H = -\Delta|e\rangle\langle e| + \frac{\Omega}{2}(|e\rangle\langle g| + |g\rangle\langle e|)$ describes a typical two-level system under drive, where $\Delta = \omega_l - \omega_0$ denotes the detuning of the driving laser $\omega_l$ from the resonance transition $\omega_0$ of the two levels and $\Omega$ represents the Rabi frequency. $L_A[\rho] \equiv A\rho A^\dagger - \frac{1}{2}\{A^\dagger A, \rho\}$. $\gamma_\phi$ means the dephasing rate associated with $|e\rangle$, and $\gamma_0$ is the effective decay rate from $|e\rangle$ to $|g\rangle$. For convenience of the following investigation, we assume $\gamma_\phi + \gamma_0 = \gamma$ with $\gamma_\phi = (1 - \alpha)\gamma$ and $\gamma_0 = \alpha\gamma$. Following the convention, we employ $\gamma/\Omega$ to quantify positions of LEPs in the phase space, where the regime before (after) the LEP is called the exact (broken) phase[34].

For our purpose, we rewrite the Lindblad master equation by a matrix of Liovillian superoperator as

$$\widehat{L}(\alpha) = \begin{pmatrix} -i\alpha\gamma & -\frac{\Omega}{2} & \frac{\Omega}{2} & 0 \\ -\frac{\Omega}{2} & -i\frac{\gamma}{2} - \Delta & 0 & \frac{\Omega}{2} \\ \frac{\Omega}{2} & 0 & -i\frac{\gamma}{2} + \Delta & -\frac{\Omega}{2} \\ i\alpha\gamma & \frac{\Omega}{2} & -\frac{\Omega}{2} & 0 \end{pmatrix}, \quad (2)$$

with the Liovillian superoperators for LEPs due to dephasing and LEPs due to decay as $\widehat{L}_\phi = \widehat{L}(\alpha = 0)$ and $\widehat{L}_0 = \widehat{L}(\alpha = 1)$, respectively. Considering $\widehat{L}(\alpha) = (1 - \alpha)\widehat{L}_\phi + \alpha\widehat{L}_0$, we solve its eigenvalues and eigenvectors. If $\Omega$ remains constant, we obtain three complex eigenenergies $E_1$, $E_2$, and $E_3$ as well as $E_4 = 0$ (See Supplementary Section I). By fixing $\Omega$ and $\alpha$, and carefully tuning $\Delta$ and $\gamma$, we find the linear variation of the second-order LEP, i.e., coalescence of two eigenstates with the same eigenenergy. In particular, when $\Delta/\Omega = \pm 1/\sqrt{8}$ and $\gamma/\Omega = \sqrt{\frac{2}{2}}/|1-2\alpha|$, three eigenstates coalesce with the same eigenenergy, known as the third-order LEP (See Supplementary Section I). We show in Fig. 1(b) the variation of the second- and third-order LEPs by adjusting $\Delta$, $\alpha$ and $\gamma$, where both the second- and third-order LEPs diverge to infinity when $\alpha = 0.5$. This is due to the fact that $E_1 = -\frac{i\gamma}{2} - \sqrt{\Delta^2 + \Omega^2}$, $E_2 = -\frac{i\gamma}{2}$ and $E_3 = -\frac{i\gamma}{2} + \sqrt{\Delta^2 + \Omega^2}$ when $\alpha = 0.5$, for which the imaginary part is completely degenerate, whereas the real part is fully separated and does not decrease with the increase of dissipation.

Our experiment is carried out by employing the internal states of a single ultracold $^{40}Ca^+$ ion confined in a surface-electrode trap (SET), as shown in Fig. 1c. This trap is a 500-$\mu$m-scale planar trap, similar to those employed previously in refs. 48–50 while more complicated. The secular frequencies of the SET are $\omega_z/2\pi = 0.65$ MHz, $\omega_x/2\pi = 1.2$ MHz and $\omega_y/2\pi = 1.59$ MHz, respectively. The confined ion stays stably above the surface of the SET by 500 $\mu$m. Under an external magnetic field of 0.51 mT directed along axial orientation, the ground state $4^2S_{1/2}$ and the metastable state $3^2D_{5/2}$ are split into two and six Zeeman energy sublevels, respectively. As plotted in Fig. 1(d), we label the ground state $|4^2S_{1/2}, m_J = +1/2\rangle$ as $|g\rangle$, the metastable state $|3^2D_{5/2}, m_J = +5/2\rangle$ as $|e\rangle$

and the auxiliary state $|4^2P_{3/2}, m_J = +3/2\rangle$ as $|p\rangle$. We perform Doppler and resolved sideband cooling of the ion to reduce the average phonon number of the z-axis motional mode below 0.1. With the 729 nm or 854 nm lasers switching on (see Fig. 1d), the three-level system reduces to an effective two-level system representing a qubit[34,51], which is used in this work to control $\gamma_0$. Moreover, to produce effective dephasing of the internal state of the trapped ion, we inject white noise signal generated by AWG to AOM, which brings phase instability to 729 nm laser. By this way, we can control the effective dephasing rate $\gamma_\phi$.

### Experimental results

To observe the LEP experimentally, we measure the evolution of quantum states governed by the Lindblad master equation and acquire the degeneracies of the eigenenergies and effective dissipation rates of the system with respect to $\alpha$. To this end, we have to measure $\gamma_0$ and $\gamma_\phi$ by varying $\Omega_{854}$ and injecting noise signals, respectively, and then measure the variation of population in $|e\rangle$. Since both $\alpha$ and $\Omega$ are fully controllable, we can carry out tomography of the system when the system turns to be in steady state by measuring the population in $|e\rangle$ in three directions, labeled as $P_{|e\rangle}^{i\in(x,y,z)}$ (See Supplementary Section II). Then we acquire the steady density matrix of the quantum system as $\rho = \sum_{i=x,y,z}(P_{|e\rangle}^i - \frac{1}{2})\sigma_i + \frac{1}{2}\mathbb{I}$ with the Pauli matrix $\sigma_i$ in single qubit as well as the eigenenergies of the system, from which we identify the LEPs at points of degeneracy.

**Movement of the second-order LEPs**. Our experiment observes the movement of the second-order LEPs at $\Delta = 0$. When $\alpha = 0$, it is positioned at $\gamma/\Omega = 4$ (see the blue point in Fig. 1a). Figure 2a, b displays simulations of real and imaginary parts of eigenenergy surfaces for the second-order LEPs with detuning $\Delta = 0$, revealing the behavior of the second-order LEPs. These second-order LEPs appear at $\gamma/\Omega = 4/|1 - 2\alpha|$, and their movement arises from the non-commutativity of the Liouvillian superoperators for LEPs due to decay and LEPs due to dephasing (i.e., $[\widehat{L}_0, \widehat{L}_\phi] \neq 0$), behaving as the competition between decay and dephasing processes. This non-commutativity of the Liouvillian superoperators implies that the eigenvalues and eigenvectors of $\widehat{L}_0$ for decay processes are distinct from those of $\widehat{L}_\phi$ for dephasing processes. Thus, the Liouvillian superoperator $\widehat{L}(\alpha) = \alpha\widehat{L}_0 + (1 - \alpha)\widehat{L}_\phi$ describes the transition processes from pure dephasing processes ($\widehat{L}_\phi$ for $\alpha = 0$) to pure decay processes ($\widehat{L}_0$ for $\alpha = 1$) by increasing $\alpha$ from 0 to 1. Such a superoperator $\widehat{L}(\alpha)$ takes new eigenvalues and eigenvectors that characterize Lindblad LEPs, incompatible with the one for either $\widehat{L}_0$ or $\widehat{L}_\phi$. Therefore, $\widehat{L}(\alpha)$ does not simply inherit the spectral properties of $\widehat{L}_0$ and $\widehat{L}_\phi$, but instead generates a unique set of eigenvalues and eigenvectors that reflect more complex dynamics and cannot be simply deduced from the individual Liouvillian superoperator $\widehat{L}_0$ or $\widehat{L}_\phi$. Besides, the Lindblad LEPs, governed by the LEPs due to dephasing ($0 \leq \alpha < 0.5$), exhibit less dissipation rates than those dominated by the LEPs due to decay ($0.5 < \alpha \leq 1$), as illustrated in Fig. 2b, since only the latter involve the energy loss.

By initializing the system in $|g\rangle$ and setting $\Omega/2\pi = 40$ kHz and $\Delta = 0$, we experimentally observe the trajectories of the movable second-order LEPs by varying $\alpha$. Figure 2c shows that the experimental results are in good agreement with the theoretical simulations, with the positions of the second-order LEPs following $\gamma/\Omega = 4/|1 - 2\alpha|$. With the increase of $\alpha$, the second-order LEPs trace a trajectory from the LEPs due to pure dephasing ($\alpha = 0$), to the Lindblad LEPs dominated by dephasing ($0 < \alpha < 0.5$), and then to the Lindblad LEPs governed by decay ($0.5 < \alpha < 1$), and finally to the LEPs due to pure decay ($\alpha = 1$). The Lindblad LEPs span two distinct regimes: one dominated by the dephasing with $0 < \alpha < 0.5$ and the other governed by the decay with $0.5 < \alpha < 1$. These two regimes are separated by a critical value of $\alpha = 0.5$, corresponding to the second-order LEP residing at $\gamma/\Omega = \infty$. At $\alpha = 0.5$, the dephasing rate and the decay rate are balanced, satisfying the condition $|\gamma_\phi - \gamma_0| = |1 - 2\alpha|\gamma = 0$ and resulting in a

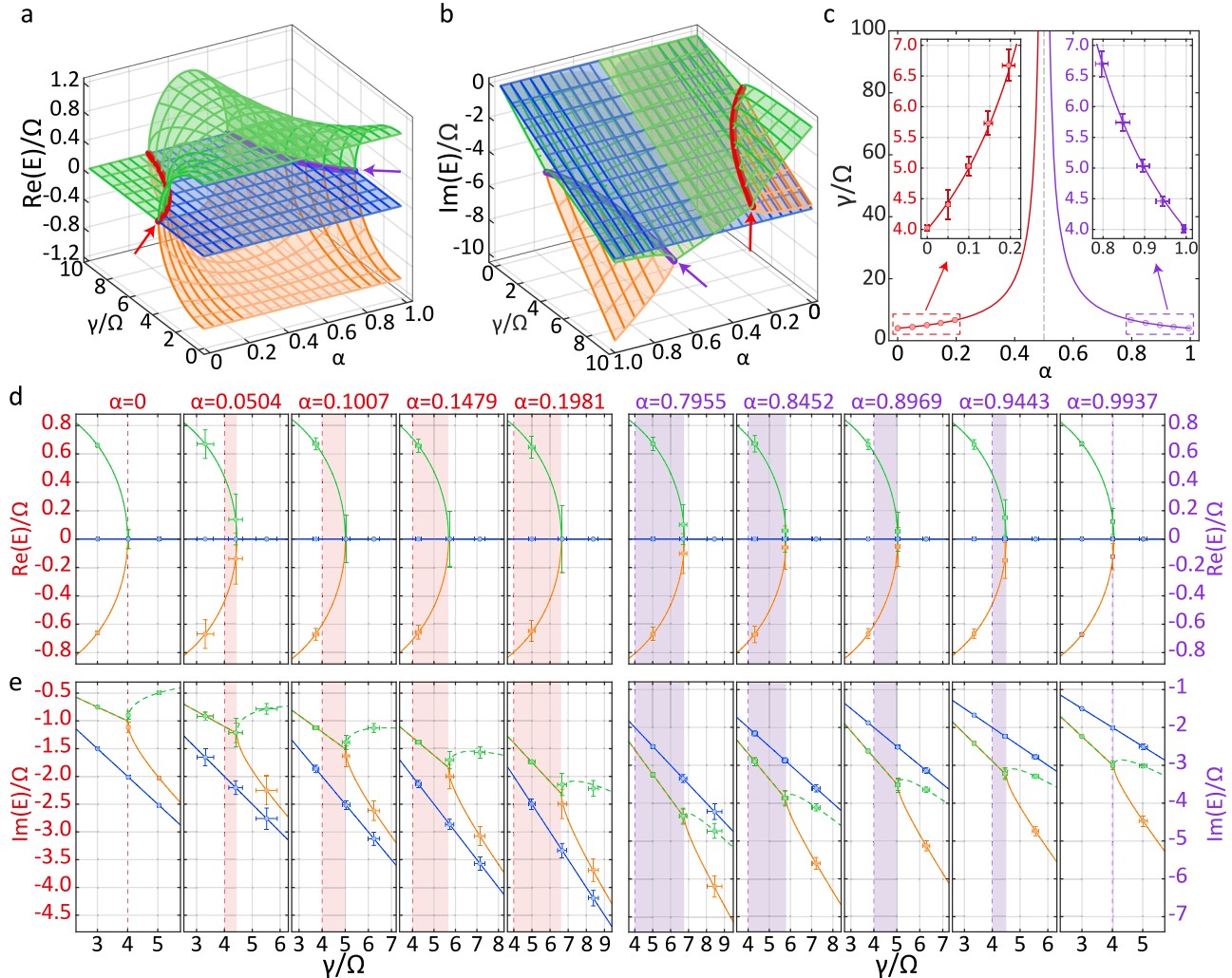

**Fig. 2 | Movement of second-order LEPs. a** The real part and **b** the imaginary part of the complex eigenenergy Riemann surfaces when $\Delta = 0$. The red and purple lines represent, respectively, the trajectories of the second-order LEP's movement dominated by dephasing and decay. Since the trajectories are largely hidden by Riemann surfaces, we draw short arrows for guiding the eye. **c** The red and purple lines are theoretical results of the trajectories of the LEP's movement. Dots with error bars stand for the experimentally observed second-order LEPs with different $\alpha$. The orange, blue and green dots with error bars represent (**d**) the real and **e** the imaginary parts of the eigenvalues acquired from experimental measurement. The lines are calculated from master equation. Red (purple) area in each panel represents the movement range of the Lindblad LEP with $\alpha$ varying from $\alpha = 0(1)$ (i.e., the LEP due to dephasing or decay, labeled by a vertical dashed line) to the value labeled on the top of the panel. $E_1$ (orange line) and $E_3$ (green line) turn to be degenerate before the LEP is reached. For clarity, we replace the green solid line by the dashed line. The error bars are standard deviation representing the statistical errors of 14 000 measurements for each data point.

counterintuitive feature that different eigenenergies $\mathrm{Re}(E)$ of $\hat{L}(\alpha = 0.5)$ share the same dissipation rate $\mathrm{Im}(E)$. Moreover, such balance implies the vanishment of second-order LEPs, merging the LEPs due to dephasing ($0 < \alpha < 0.5$) and decay ($0.5 < \alpha < 1$) at $\gamma/\Omega = \infty$. Conversely, when the dephasing rate and the decay rate differ, the enhanced difference of the dissipation rates (i.e., $|\gamma_\phi - \gamma_0| = |1 - 2\alpha|\gamma > 0$) will shift the second-order LEPs toward $\gamma/\Omega = 4$ at the extreme values of $\alpha = 0$ and $\alpha = 1$. Therefore, Fig. 2c illustrates this trajectory that, as $\alpha$ increases, the value of Lindblad LEPs first rises from $\gamma/\Omega = 4$ to $\gamma/\Omega = \infty$ (see the lefthand side region), and then decreases from $\gamma/\Omega = \infty$ to $\gamma/\Omega = 4$ (see the righthand side region).

To scrutinize the movement of the LEPs, we present separate panels for different values of $\alpha$ in Fig. 2d, e, exhibiting the eigenvalues of the superoperator $\hat{L}(\alpha)$. The eigenenergy $\mathrm{Re}(E_2)$ (in blue) remains zero with the dissipation rate $\mathrm{Im}(E_2)$ linear in $\gamma$, while the other two eigenvalues $E_1$ (in orange) and $E_3$ (in green) exhibit distinct behavior. When $\gamma/\Omega < 4/|1 - 2\alpha|$, $\mathrm{Re}(E_1)$ and $\mathrm{Re}(E_3)$ are completely separate, but their dissipation rates are completely degenerate, indicating in exact

phase regime. When $\gamma/\Omega > 4/|1 - 2\alpha|$, eigenenergies $\mathrm{Re}(E_1)$ and $\mathrm{Re}(E_3)$ become degenerate, taking the value of zero, but their dissipation rates are completely separate, implying in broken phase regime. The transition at $\gamma/\Omega = 4/|1 - 2\alpha| = \infty$, dividing the exact and broken phase regimes, highlights the second-order LEPs of the eigenvalues for $\Delta = 0$. The characteristic separations and degeneracies in eigenenergies with respect to dissipation rates, associated with $E_1$ and $E_3$ but not with $E_2$, remind us that these LEPs are of the second order.

These movable second-order LEPs reveal a fundamental result that the non-commutativity of the Liouvillian superoperators $\hat{L}_\phi$ and $\hat{L}_0$ drives the movement of the second-order LEPs, governs their positions at $\gamma/\Omega = 4/|1 - 2\alpha|$, and controls the transition points between the exact-phase and broken-phase regimes. In physics, as the non-commutative Liouvillian superoperators take different eigenvalues and eigenstates, the competition of these Liouvillian superoperators results in the movement of the Lindblad LEPs. This highlights that the EP movement is a universal feature of non-Hermitian systems, stemming from the ubiquitous presence of non-commutative operators. Consequently,

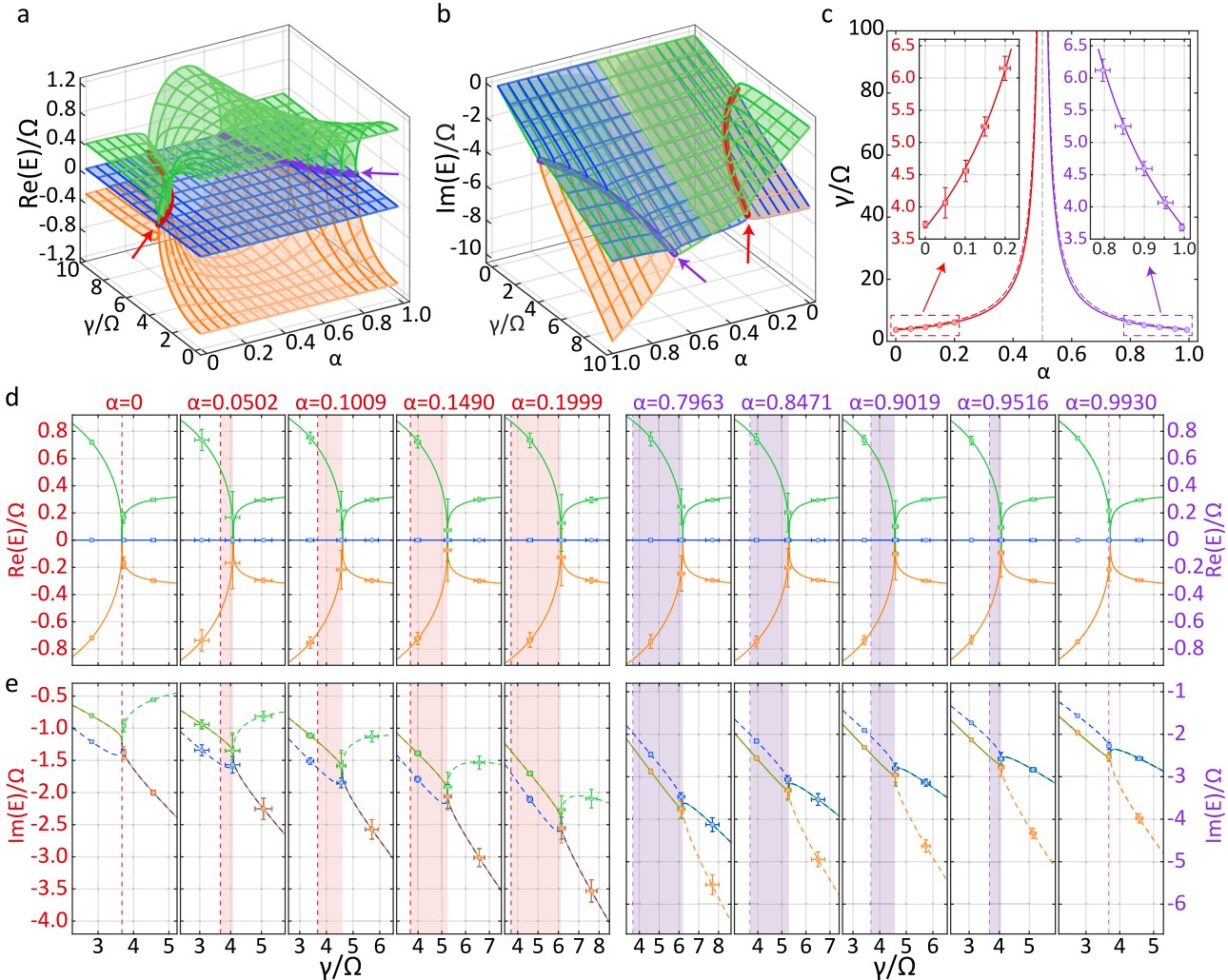

**Fig. 3 | Movement of third-order LEPs. a** The real part and **b** the imaginary part of the complex eigenenergy Riemann surfaces when $\Delta/\Omega = 1/\sqrt{8}$. The red and purple lines represent, respectively, the trajectories of the third-order LEP's movement dominated by dephasing and decay. Since the trajectories are largely hidden by Riemann surfaces, we draw short arrows for guiding the eye. **c** The red and purple lines are theoretical calculation of the trajectories of the LEP's movement. Dots with error bars stand for the experimentally observed third-order LEPs with different $\alpha$. In order to compare with the second-order LEPs, we added red and purple dashed lines when $\Delta = 0$. The orange, blue and green dots with error bars are (**d**) the real and **e** the imaginary parts of the eigenvalues acquired from experimental

measurement. The lines are calculated from master equation. Red or purple area in each panel represents the movement range of the Lindblad LEP with $\alpha$ varying from $\alpha = 0$ (i.e., the LEPs due to dephasing, labeled by a vertical dashed line) to the value labeled on the top of the panel. $E_1$ (orange) and $E_3$ (green) are degenerate before reaching LEP, while after crossing the LEP, $E_1$ (orange) and $E_2$ (blue) become degenerate when $\alpha < 0.5$, and $E_2$ (blue) and $E_3$ (green) become degenerate when $\alpha > 0.5$. For clarity, we replace some overlapped solid lines by dashed lines. The error bars represent standard deviation, i.e., the statistical errors of 14 000 measurements for each data point.

the non-commutativity offers a valuable insight into a fundamental question in non-Hermitian system for the mechanism behind the EP movement from the HEP[2,3] to LEP[36,37,39] when $\Delta = 0$[32,33].

**Movement of the third-order LEPs.** Owing to the presence of quantum jumps, adjusting the detuning from zero to non-zero may reveal third-order LEPs at the intersectional points of second-order Liouvillian exceptional lines on the three eigenenergy Riemann surfaces (see the red points and purple curves in Fig. 1b). These Liouvillian exceptional lines can also be moved by adjusting $\alpha$. When $\alpha = 0$ and $\Delta = \pm\Omega/\sqrt{8}$, the third-order LEPs are at $\gamma/\Omega = \sqrt{27/2} < 4$, as indicated by the red points in Fig. 1a, b.

Figure 3a, b presents simulations of real and imaginary parts of eigenenergy for the third-order LEPs, exhibiting the movement of the third-order LEP located at $\Delta = \Omega/\sqrt{8}$ and $\gamma/\Omega = \sqrt{27/2}/|1 - 2\alpha|$. This behavior arises from the non-commutativity of the Liouvillian superoperators $\hat{L}_\phi$ and $\hat{L}_0$. Different from the second-order LEPs, which are

associated only with $E_1$ and $E_3$, the third-order LEPs involve the eigenvalues $E_1$, $E_2$ and $E_3$. Therefore, all the eigenvalues work as functions of $\alpha$, similar to those of the second-order LEPs. Also similar to the second-order LEPs at $\alpha = 0.5$, the third-order LEPs own the eigenenergies Re($E$) sharing identical dissipation rates Im($E$) (see Fig. 3b). However, no degeneracy for the eigenenergies Re($E$) occurs at $\Delta/\Omega = 1/\sqrt{8}$, see Fig. 3a.

Figure 3c presents our experimental observations of the trajectories of the movable third-order LEPs, in which the values of the third-order LEPs are lower than those of the second-order LEPs. This difference arises due to the fact that the third-order LEPs are localized at $\gamma/\Omega = \sqrt{27/2}/|1 - 2\alpha|$, whereas the second-order LEPs are positioned at $\gamma/\Omega = 4/|1 - 2\alpha|$. As a result, increasing $\alpha$ can first move the third-order LEPs from $\gamma/\Omega = \sqrt{27/2}$ to $\gamma/\Omega = \infty$ for $0 \leq \alpha < 0.5$, and then shift them back from $\gamma/\Omega = \infty$ to $\gamma/\Omega = \sqrt{27/2}$ for $0.5 < \alpha \leq 1$.

Similar to the second-order LEPs, the third-order LEPs exhibit some interesting properties. According to $|1 - 2\alpha| = |\gamma_\phi - \gamma_0|/\gamma$, the

third-order LEPs share the same positions at $\gamma/\Omega = \sqrt{27/2}$ for both the pure dephasing with $\alpha = 0$ and the pure decay with $\alpha = 1$, while corresponding to distinctly different eigenenergies and eigenstates. When $0 < \alpha < 1$, the third-order Lindblad LEPs are characterized by the movement resulting from the fact that the Lindblad superoperators for dephasing $\hat{L}_\phi = \hat{L}(\alpha = 0, \Delta/\Omega = \pm 1/\sqrt{8})$ and for decay $\hat{L}_0 = \hat{L}(\alpha = 1, \Delta/\Omega = \pm 1/\sqrt{8})$ do not commute. As a result, when the dephasing rate equals to the decay rate, i.e., $\gamma_\phi = \gamma_0$, the LEP shifts to $\gamma/\Omega = \infty$. In this case, the energy spacing of the eigenstates is uniform and independent of dissipation, see Eq. (2) with $\Delta \neq 0$.

The presence of the third-order LEPs underscores how quantum jump provides an additional degree of freedom in a two-level quantum system. This phenomenon results from that quantum jumps, inherent to the Lindblad formalism, couple the coherent dynamics to dissipation processes, expanding the dimensionality of Liouvillian space. In contrast, the third-order Hamiltonian EPs typically require three or more modes due to the expanded dimensionality of the Hilbert space. This indicates that, although non-Hermitian two-level Hamiltonian is limited to second-order EPs, quantum jumps enrich the Liouvillian space, enabling the existence of third-order LEPs.

Figure 3d, e shows variations of eigenvalues acquired both theoretically and experimentally. In Fig. 3d, all eigenenergies $\mathrm{Re}(E_1)$, $\mathrm{Re}(E_2)$ and $\mathrm{Re}(E_3)$ are completely separate due to the presence of the detuning $\Delta/\Omega = 1/\sqrt{8}$, in which the exception is for the ones localized at the third-order LEPs ($\gamma/\Omega = \sqrt{27/2}/|1 - 2\alpha|$). We find that the eigenenergies near the third-order LEPs vary more steeply than those for the second-order LEPs. Consequently, the third-order LEP might be utilized to enhance measurement accuracy, and the movement of the third-order LEPs suggests a novel approach to designing high-sensitivity precision measurement by adjusting $\alpha$ to shift the third-order LEPs. Moreover, these third-order LEPs appear due to quantum jumps, and thus lead to counterintuitive result that quantum jumps, typically associated with decoherence, can improve the high-sensitivity precision measurement through the high-order LEPs within the enriched Liouvillian space as treated previously for high-order HEPs in multi-level systems[11,43,52].

In contrast, the presence of degenerations in dissipation rates in Fig. 3e indicates that $\alpha = 0.5$ divides the system into two working regimes of the Lindblad LEPs, i.e., governed by LEPs due to dephasing ($0 \leq \alpha < 0.5$), and dominated by LEPs due to decay ($0.5 \leq \alpha < 1$). In the case of $0 \leq \alpha < 0.5$, when $\gamma/\Omega < \sqrt{27/2}/|1 - 2\alpha|$, dissipation rates $\mathrm{Im}(E_1)$ in orange and $\mathrm{Im}(E_3)$ in green are degenerate, separated from $\mathrm{Im}(E_2)$ in blue. When $\gamma/\Omega = \sqrt{27/2}/|1 - 2\alpha|$, all dissipation rates degenerate, i.e., the typical characteristic of third-order LEP. When $\gamma/\Omega > \sqrt{27/2}/|1 - 2\alpha|$, two degenerated dissipation rates (i.e., $\mathrm{Im}(E_1)$ in orange and $\mathrm{Im}(E_2)$ in blue) are separated from $\mathrm{Im}(E_3)$ in green. These features indicate that the third-order LEP emerges from the intersection of second-order Liouvilian exceptional lines for $E_1 = E_3$ and $E_1 = E_2$. Similarly, in the case of $0.5 \leq \alpha < 1$, when $\gamma/\Omega < \sqrt{27/2}/|1 - 2\alpha|$, two dissipation rates (i.e., $\mathrm{Im}(E_1)$ in orange and $\mathrm{Im}(E_3)$ in green) are degenerate, distinct from $\mathrm{Im}(E_2)$ in blue; at $\gamma/\Omega = \sqrt{27/2}/|1 - 2\alpha|$, all dissipation rates degenerate, characterized by the third-order LEPs. When $\gamma/\Omega > \sqrt{27/2}/|1 - 2\alpha|$, two degenerated dissipation rates (i.e., $\mathrm{Im}(E_2)$ in blue and $\mathrm{Im}(E_3)$ in green) are distinct from $\mathrm{Im}(E_1)$ in orange. Notably, as only the decay process involves energy loss, with the increase of $\alpha$, the energy loss will rise, resulting in the increase of dissipation rates in Fig. 3e.

Finally, evolutions around a single third-order LEP manifest to be topologically non-trivial. By independently varying $\alpha$, the system can switch between topologically trivial and non-trivial phases, depending on whether the evolution encircles a single third-order LEP or not. This tunability of topological properties suggests that the movement of a single third-order LEP could be potentially useful in demonstrating non-Hermitian topological properties[9,15,17,18,46,47,53,54] and topological dynamics[7,8,37,40] resulting from quantum jumps[30,31]. Besides, the third-order LEP occurs at $\Delta \neq 0$, resulting from quantum jump, unlike the third-order HEP at $\Delta = 0$, which neglect quantum jump terms[55].

## Discussion

In summary, we have demonstrated the emergence and movement of the second- and third-order LEPs in a two-level quantum system, governed by Liouvillian superoperators. By continuously tuning the characteristic parameters from the LEPs due to pure dephasing to the LEPs due to pure decay, we have observed movable LEPs in the phase space, driven by non-commutativity of the dephasing and decay Liouvillian superoperators. Remarkably, the third-order LEPs, enabled by quantum jumps, expand the Liouvillian space beyond the limitations of a two-level non-Hermitian Hamiltonian. Compared to the second-order LEPs, the third-order LEPs own enhanced eigenvalue variation, taking potential in improving precision measurement sensitivity and enabling topological phase switching. In this context, our results reveal that quantum jumps, typically associated with decoherence, could counterintuitively improve non-Hermitian precision measurement as mentioned before[11,13,14] and also enable topological control using the third-order LEPs[15-19,56,57], thus bridging fundamental non-Hermitian physics to practical applications[4,58] across fundamental physics, engineering, and quantum technology.

Achieving higher sensitive measurement on specific parameters based on the third-order LEPs is the focus of our next experimental work. By a point-to-point comparison between Figs. 2d, e and 3d, e, we find that, although the eigenenergies near the third-order LEPs change more sharply than those near the second-order LEPs, no error bars with significantly different sizes exist between the two figures, indicating no more imperfection in experimental operations involving third-order LEPs than second-order LEPs. This suggests that trapped-ion quantum sensors hold promise for highly sensitive detection, particularly when operating around third-order LEPs.

Moreover, we have noticed an interesting work[28] investigating theoretically the EPs relevant to non-Markovian dynamics. In comparison to Markovian cases, the non-Markovian effects can effectively increase the dimensionality of the associated non-Hermitian super-operators, helping for searching higher-order EPs. However, it is hard to demonstrate this idea in our experimental setup. If we would like to experimentally achieve a LEP with non-Markovian dynamics, we have to introduce strong dissipation in our system. Unfortunately, our effective two-level system, originating from a three-level system as shown in Fig. 1d, could not reach the strong decay since the strength of 854nm laser is required to be much smaller than the decay rate of the excited state (i.e., the auxiliary state $|P\rangle$)[51]. Nevertheless, this theoretical proposal provides an alternative way to finding higher-order LEPs. Therefore, how to produce a non-Markovian LEP experimentally is something worth trying hard in the near future.

## Methods

### Measurement of $\gamma_0$ and $\gamma_\phi$

In our experiment, we define $\gamma_0$ as the effective decay rate induced by the 854nm laser, and $\gamma_\phi$ as the dephasing rate resulting from phase modulation of the 729nm laser using white noise. These two parameters are calibrated prior to each experimental run, and their dependencies on the power of the 854nm laser and the amplitude of the applied white noise are systematically investigated. In the following, we specify the procedures for measuring $\gamma_0$ and $\gamma_\phi$, as well as their relations to experimentally controlled parameters.

Figure 4 presents the measured dependence of the effective decay rate $\gamma_0$ on the 854nm laser power in our experimental system. The measurement procedure is as follows. After Doppler cooling and sideband cooling are executed, the ion is prepared near its motional ground state. The internal state is initialized to $|g\rangle \equiv |4^2S_{1/2}, m_J = +1/2\rangle$ using an optical pumping scheme that involves the states $|4^2S_{1/2}, m_J = -1/2\rangle$ and $|3^2D_{5/2}, m_J = +3/2\rangle$, assisted by both the

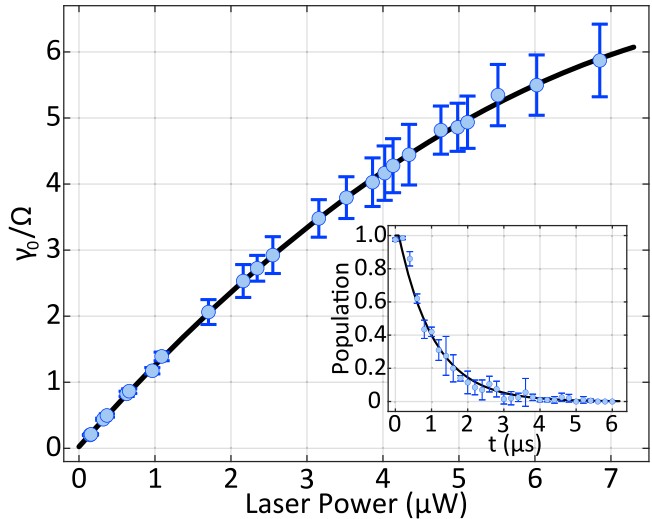

**Fig. 4 | Measured dependence of the effective decay rate $\gamma_0$ on the 854nm laser power in our experimental system.** The data are well described by a cubic polynomial fit of the form $ax^2 + bx + c$, with a coefficient of determination $R^2 = 0.9996$. The fitted coefficients (95% confidence intervals) are: $a = -0.0643$, $b = 1.30$, and $c = 0.244$. The inset shows a representative decay curve measured at the 854-nm laser power of 3.86 $\mu$W. Each data point is obtained by averaging over 200 experimental repetitions with error bars indicating standard deviation.

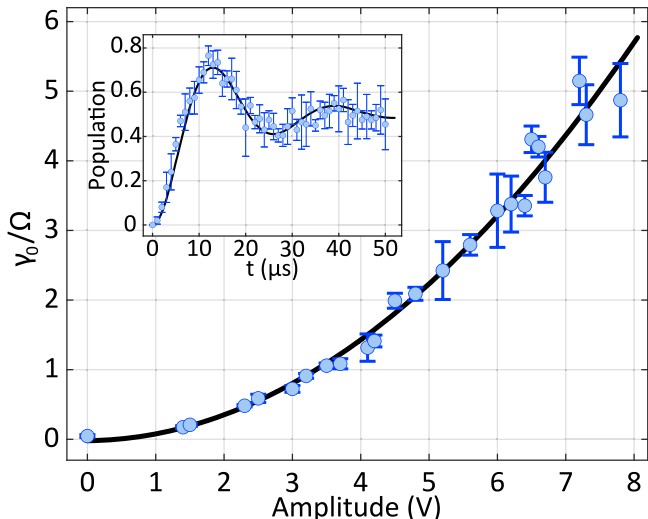

**Fig. 5 | Measured dependence of the dephasing rate $\gamma_\phi$ on the amplitude of the applied white noise, specified in peak-to-peak voltage (Vpp).** The data are well described by a cubic polynomial fit of the form $ax^2 + bx + c$, with the determination coefficient of $R^2 = 0.9800$. The fitted coefficients (95% confidence intervals) are: $a = 0.0882$, $b = 0.00940$, and $c = -0.0201$. The left and right insets display representative fits of the dephasing-induced population decay measured at white noise amplitudes of 3.5 V. Each data point is obtained by averaging over 200 experimental repetitions with error bars indicating standard deviation.

729nm and 854nm lasers. A 729nm $\pi$-pulse, calibrated to the Rabi frequency of $2\pi \times 40$ kHz, is then applied to coherently transfer the population to the excited state $|e\rangle \equiv |3\,^2D_{5/2}, m_J = +5/2\rangle$.

Following the state preparation, the 854 nm laser is applied with varying durations, introducing an effective decay rate $\gamma_0$ to the system during each time interval. After each interval, the remaining population in the metastable D state is measured using the electron-shelving technique. The population decay exhibits an approximately exponential dependence on time. By fitting the decay curves at different laser powers, we extract the corresponding effective decay rates, which are denoted by $\gamma_0$ throughout this work.

Figure 5 presents the measured dependence of the dephasing rate $\gamma_\phi$ on the amplitude of the applied white noise in our experimental system. Before describing the measurement procedure, we first explain how dephasing is implemented in our setup.

Dephasing is induced by phase modulation of the 729 nm laser using a single-pass AOM, where the negative first-order diffracted beam at a fixed frequency of 80 MHz is utilized. The AOM is driven by a sinusoidal signal from a function generator, with the phase modulation depth set to 180 degrees. To introduce noise, the function generator operates in external modulation mode, in which the white noise signal generated by an AWG is applied. The AWG produces white noise with a fixed bandwidth of 100 kHz, and its amplitude is varied to realize different dephasing strengths.

With this method, we are able to introduce tunable dephasing into the system, and then perform a careful calibration to extract the corresponding dephasing rate $\gamma_\phi$. The detailed measurement procedure is as follows.

After Doppler cooling and sideband cooling are executed, the optical pumping is applied to prepare the ion near its motional ground state, with the internal state of the ion initialized to $|g\rangle$. Then we apply the 729nm laser with varying durations to be in resonance with the carrier transition between $|g\rangle$ and $|e\rangle$, during which phase modulation is applied to introduce dephasing.

We measure the population in the metastable D state as a function of the interaction time. The resulting dynamics exhibits Rabi-like oscillations with a rapidly decaying amplitude due to the introduced dephasing. By fitting the measured population evolution to a theoretical curve – acquired using the master equation [i.e., Eq. (1) in the main text] that includes only the dephasing term – we extract the corresponding dephasing rate $\gamma_\phi$. As shown in the figure, the error bars increase with stronger dephasing. This is because, at higher dephasing rates, the Rabi-like curves become more similar in shape, resulting in greater uncertainty in the fitting process.

**Tomography of states**

An arbitrary state of the single qubit can be expressed as

$$\rho = \frac{1}{2}\left(\mathbb{I} + \sum_{i=x,y,z} S_i \sigma_i\right),$$

where $\sigma_x$, $\sigma_y$, and $\sigma_z$ are Pauli matrices, and the Stokes parameters are defined as $S_i = \text{Tr}(\sigma_i \rho)$. Full reconstruction of the density matrix requires independent measurements of all three $S_i$ components.

Experimentally, each Stokes parameter is obtained by performing projective measurements in the eigenbasis of the corresponding Pauli operator. Specifically, $S_x = P_e^x - P_g^x$, $S_y = P_e^y - P_g^y$, and $S_z = P_e^z - P_g^z$, where $P_{e,g}^i$ denote the populations in the eigenstates of $\sigma_i$. The eigenstates for the $\sigma_x$ basis are $|g\rangle^x = (|e\rangle - |g\rangle)/\sqrt{2}$ and $|e\rangle^x = (|e\rangle + |g\rangle)/\sqrt{2}$; for the $\sigma_y$ basis, $|g\rangle^y = (|e\rangle - i|g\rangle)/\sqrt{2}$ and $|e\rangle^y = (|e\rangle + i|g\rangle)/\sqrt{2}$; and for the $\sigma_z$ basis, the states are simply the computational basis states $|g\rangle^z = |g\rangle$ and $|e\rangle^z = |e\rangle$.

In our experimental system, the fidelity of state preparation and measurement can approach unity. This implies that for each $i$, $P_e^i + P_g^i \approx 1$, and therefore it is sufficient to measure only $P_e^i$ to reconstruct the full density matrix. Accordingly, the density matrix can be rewritten as

$$\rho = \frac{1}{2}\left(\mathbb{I} + \sum_{i=x,y,z}(2P_e^i - 1)\sigma_i\right) = \frac{1}{2}\mathbb{I} + \sum_{i=x,y,z}\left(P_e^i - \frac{1}{2}\right)\sigma_i.$$

We now describe how each $P_e^i$ parameter is measured. For all operations and measurements, following Doppler cooling, sideband cooling and optical pumping, we employ an AWG to precisely control the relative phase and Rabi frequency of the 729nm laser.

To measure $P_e^z$, we directly use the electron-shelving technique, where the population in the metastable D state corresponds to $P_e^z$. For $P_e^x$ and $P_e^y$, additional basis rotations are required before the measurement. Specifically, a $\pi/2$ rotation around the $y$-axis is applied to measure $P_e^x$, and a $\pi/2$ rotation around the $x$-axis is applied to measure $P_e^y$. These operations correspond to $R(\pi/2, \pi/2)$ and $R(\pi/2, 0)$ respectively. Here, $R(\theta, \phi)$ denotes a rotation by angle $\theta$ with phase $\phi$ implemented via the 729nm laser. After these rotations, the populations in the D state directly yield the values of $P_e^x$ and $P_e^y$.

## Data availability

Source data related to the figures in this paper are available from the Figshare repository at https://doi.org/10.6084/m9.figshare.30343429.

## Code availability

The codes used for the numerical calculations are available from the corresponding authors upon request.

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

## Acknowledgements

JQZ thanks for Jing-Xin Liu and Prof. Li Ge for helpful discussion and Dr. Jie Zhao for numerical simulation with Matlab. This work was supported by the National Key R&D Program under Grant (No. 2024YFE0102400, H.J.), by Quantum Science and Technology-National Science and Technology Major Project under grant (No. 2023ZD0300400, X.M.C.), by National Natural Science Foundation of China under Grant (Nos. 12534020,M.F.; 12421005,H.J.; U21A20434,M.F.;11935006,H.J.), by Guangdong Provincial Quantum Science Strategic Initiative under Grant (No. GDZX2305004,M.F.), by Natural Science Foundation of Wuhan under Grant (No. 2024040701010063,F.Z.), by Hunan Major Sci-Tech Program under Grant (No. 2023ZJ1010,H.J.), by Nansha Senior Leading Talent Team Project under Grant (No. 2021CXTD02,M.F.), and by the Special Project for Research and Development in Key Areas of Guangdong Province under Grant (No. 2020B0303300001,M.F.).

## Author contributions

J.Q.Z., L.C., and M.F. conceived the idea and designed the research. Z.-Z.W. and P.-D.L. carried experiments with help from T.-H.C., Y.-Z.D., S-.Q.D., Y-.Q.W., and J.L. Theoretical background and simulations were provided by Z-.Z.W. and J-.Q.Z. with help from J-.W.W., Q.Y., X-.M.C., and H.J. All authors contributed to the discussions and the interpretations of the experimental and theoretical results. Z-.Z.W., J-.Q.Z., and M.F. wrote the manuscript with inputs from all authors.

## Competing interests

The authors declare no competing interests.
