## [Transparent Peer Review file · Nature Communications]

Experimental Witness of Quantum Jump Induced High-Order Liouvillian Exceptional Points

Corresponding Author: Dr Jian-Qi Zhang

Version 0:

Reviewer comments:

Reviewer #2

(Remarks to the Author)

Wu et al. have experimentally studied higher-order exceptional points (EPs) of the Liouvillian superoperator in a trapped-ion-based two-level system. By varying the ratio of pure dephasing to energy decay, they also observed the movement and transition of both second- and third-order Liouvillian EPs in the parameter space. The experiments were well-designed, and the results are solid. I could recommend this manuscript for publication in Nature Communications after the following comments have been addressed.

My primary concern is that although this work shows the movements of both second- and third-order LEPs, it does not sufficiently address the significance of these observations. The authors conclude that "Our results also reveal that quantum jumps, typically associated with decoherence, could counterintuitively improve non-Hermitian precision measurement," but this claim requires more proof.

Adding an analysis of the sensing performance would significantly increase the paper's impact. I suggest the authors conduct a detailed analysis of the sensing performance based on their current results to answer key questions, such as:

- At what value of the parameter α does the system exhibit the strongest response?
- Does EP3 offer an advantage than EP2 when considering both the eigenvalue splitting and the error bar?

Presentation

- The word "black" appears throughout the manuscript without relevant context. The surrounding sentences also need to be rephrased. For instance, the sentence "a single second-order HEP can be achieved by blackits Liouvillian superoperator" needs to be corrected for clarity.
- The terms "decay LEPs" and "dephasing LEPs" could be confusing in the context. I suggest using more precise terms like "LEP due to decay" or "LEP due to dephasing," or other proper notations.

Reviewer #3

(Remarks to the Author)

The article by Wu et al. is an experimental investigation of Liouvillian exceptional points (LEPs) in an ultracold trapped ion system. The main idea is to use decay and dephasing processes in an effective two-level atom to probe EPs in phase space through tomography. While there are a few works on the experimental demonstration of LEPs, the paper claims to be the first one to demonstrate a third-order Liouvillian EP. Indeed, the only other work on the topic that I have seen (ref. 55) discusses a third-order Liouvillian EP in the context of no-jump dynamics, while here, quantum jumps are part of the picture. This is quite relevant for many directions that research on EPs has taken in the recent years. Moreover, their experiment shows a nice interplay between decay and dephasing rates to control the location of the EP, which I believe is also a new advance. Furthermore, the article is generally clearly written, except that there are too many typographical errors in the present version.

Although the experimental techniques used in the work are already known, overall, I believe that the main advance of this paper is significant enough to warrant publication with high visibility. Therefore, I recommend publication in Nature

Communications.

There are some issues that need the authors' attention.

1. To my knowledge, Ref. [N. Hatano, *Molecular Physics*, 117(15–16), 2121–2127] is the first one to discover a third order EP in the Liouvillian of a driven two level system, which the article does not cite. To my understanding the model used in the article is essentially identical to Hatano's.
2. It isn't entirely clear from the text why Fig. 3 has very few data points. Can the authors explain this limitation?
3. Could the authors comment on how their results compare with those in ref. 28, which observes a third—order LEP, but with non-Markovian dynamics?
4. Fig. 3 a and b are not understandable in their present form. Please also mark the EPs where needed.
5. Hybrid-LEPs have a different meaning in recent literature (ref. 33) than what the the authors mean. The authors mean a hybrid between decay and Dephasing LEPs, while the well-known Ref. 33 introduces hybrid EPs between Hamiltonian and Liouvillian EPs. It is also strange to call Eq. (1) a hybrid Lindblad equation or (2) a hybrid Liouvillian because (1) is an ordinary Lindblad equation, just with two distinct processes.

Version 1:

Reviewer comments:

Reviewer #2

(Remarks to the Author)

The authors have successfully addressed my comments. Therefore I can recommend this manuscript to be published in Nature Communications.

Reviewer #3

(Remarks to the Author)

I thank the authors for considering my comments. I believe the updated manuscript is now suitable for publication in Nature Communications.

List of Main Changes

1. Following the comments of the 3rd Reviewer, we have replotted Figs. 2 and 3, and the captions are also updated.
2. Following the comments of the two Reviewers, we have added a reference and largely rewritten the paragraphs in Discussion and Conclusion. We have also complemented a new section, i.e., Section III, in Supplementary Materials.
3. Following the comments of the 3rd Reviewer, we have replaced “hybrid LEPs” in the original version of the manuscript by “Lindblad LEPs” throughout the revised manuscript.

All the revisions are highlighted in the pdf files “highlightmaintext.pdf” and “highlightSM.pdf”.

Reply to the Reviewer #2

We thank you very much for your positive assessment of our work, particularly for the statement *"The experiments were well-designed, and the results are solid"*. We have carefully improved the manuscript according to your comments/suggestions. A one-by-one response is given below.

Comment#1

Although this work shows the movements of both second- and third-order LEPs, it does not sufficiently address the significance of these observations. The authors conclude that "Our results also reveal that quantum jumps, typically associated with decoherence, could counterintuitively improve non-Hermitian precision measurement," but this claim requires more proof.

Our Response:

We appreciate your careful review and for reminding us of this issue.

From the context of the sentence "Our results also reveal that quantum jumps, typically associated with decoherence, could counterintuitively improve non-Hermitian precision measurement", we would like to express that, it is the quantum jumps that enrich the Liouvillian space, enabling the existence of third-order Liouvillian exceptional points (LEPs), and these third-order LEPs induced by quantum jumps could help for improving the measurement sensitivity in comparison to the ordinary second-order LEPs.

The proof of this claim could be simply found from the comparison between Fig. 2(d,e) and Fig. 3(d,e) that our experimentally measured eigenvalues in the latter have steeper variations with respect to γ/Ω , indicating a better sensitivity to the characteristic parameters to be measured. In particular, a point-to-point comparison between the two figures reveals no error bars with significantly different sizes. As such, we consider that the third-order LEPs help achieve better measurement sensitivity than the second-order LEPs. Moreover, to our knowledge, there has been no report so far to specifically demonstrate better measurement sensitivity to a characteristic parameter based on the third-order LEP. This is actually the near-future work we are planning.

To make this part more clarified, we have rewritten some sentences of the Discussion and Conclusion part in our revision.

Comment#2

Adding an analysis of the sensing performance would significantly increase the paper's impact. I suggest the authors conduct a detailed analysis of the sensing performance based on their current results to answer key questions, such as:

-At what value of the parameter α does the system exhibit the strongest response?

-Does EP3 offer an advantage than EP2 when considering both the eigenvalue splitting and the error bar?

Our Response:

We sincerely appreciate your insightful comment. This comment raises two questions, both of which are relevant to Comment #1.

For the first question, we have already plotted in Figs 2 and 3 the variations of γ/Ω with respect to α (panel c) and the eigenvalues with respect to γ/Ω (panels d and e). From these panels, we can straightforwardly draw the relation of the real part of the eigenvalues to α , see Figure R1 below. As we approach the third-order LEPs, the slopes of the real part of the eigenvalue derivatives $\mathbf{d}(\text{Re}[\mathbf{E}])/\mathbf{d}\alpha$ become steeper, indicating a stronger response to the signal being detected. For the system operating at $\gamma/\Omega = 4.6$, the positions of the third-order LEPs are given by $\alpha=0.1006$ and $\alpha=0.8994$, according to Eq. (S4). Considering the experimental imperfections as demonstrated in Fig. 3, we deem that $\alpha=0.1009$ and $\alpha=0.9019$ are the optimal points in our experiment to exhibit the strongest response.

Now we answer the second question. Comparing Fig. 2(d) with Fig. 3(d), we observe that the eigenenergies near the third-order LEPs change more sharply than those near the second-order LEPs as γ/Ω varies. This suggests a higher sensitivity to the parameters being measured. Meanwhile, no error bars with significantly different sizes are found in the point-to-point comparison between the two figures. As such, we consider that the third-order LEPs help present better measurement sensitivity than the second-order LEPs.

Figure R1. Variation of the real part of the eigenvalues vs α , where γ/Ω and Δ/Ω are fixed to be 4.6 and $1/\sqrt{8}$, respectively. In this case, the third-order LEPs are located at $\alpha=0.1006$ due to dephasing dominated and at $\alpha=0.8994$ due to decay dominated, as labeled by the vertical red and purple dashed lines, respectively.

Following your suggestion, we have complemented some sentences in Discussion and Conclusion part to mention the performance of an ionic quantum sensor based on the third-order LEPs. Also, a section, i.e., Section III, is added in Supplementary Materials to justify the best performance of the ionic quantum sensor involving the third-order LEPs.

Comment#3

The word "black" appears throughout the manuscript without relevant context. The surrounding sentences also need to be rephrased. For instance, the sentence "a single second-order HEP can be achieved by blackits Liouvillian superoperator" needs to be corrected for clarity.

Our Response:

We apologize for these wrongly appeared words "black", which are resulted from the command $\backslash\text{color}\{\text{black}\}$ in the tex file of our manuscript. No such wrong information was found in compiling the tex file in our personal computers before submission of the manuscript.

To avoid this problem happening again in this resubmission, we have deleted all the commands $\backslash\text{color}\{\text{black}\}$ in the tex file of our revised manuscript.

Comment#4

The terms "decay LEPs" and "dephasing LEPs" could be confusing in the context. I suggest using more precise terms like "LEP due to decay" or "LEP due to dephasing," or other proper notations.

Our Response:

We accept your valuable suggestion and replace the terms "decay LEPs" and "dephasing LEPs" by "LEPs due to decay" and "LEPs due to dephasing", respectively, or other notations throughout the manuscript.

Reply to the Reviewer #3

We thank you very much for your positive assessment of our work, particular for the statement *“While there are a few works on the experimental demonstration of LEPs, the paper claims to be the first one to demonstrate a third-order Liouvillian EP. Moreover, their experiment shows a nice interplay between decay and dephasing rates to control the location of the EP, which I believe is also a new advance”*. We have responded to your comments one-by-one as below, and improved the manuscript according to the comments/suggestion.

Comment#1

To my knowledge, Ref. [N. Hatano, Molecular Physics, 117(15–16), 2121–2127] is the first one to discover a third order EP in the Liouvillian of a driven two level system, which the article does not cite. To my understanding the model used in the article is essentially identical to Hatano’s.

Our Response:

Many thanks for bringing this reference to our attention. We have cited it in the revised version of our manuscript as Ref. [42].

Comment#2

It isn’t entirely clear from the text why Fig. 3 has very few data points. Can the authors explain this limitation?

Our Response:

We appreciate your critical reading of our manuscript.

We guess that the “very few data points” you mentioned refer to panels d and e. These two panels demonstrate our experimental measurements of eigenenergies and dissipation rates at third-order LEPs, in which uncertainties (i.e., error bars) become larger and larger when approaching the LEPs although the measurement values fit very well with the theoretical prediction. To demonstrate our measurement values more clearly, we have just plotted very few

data points in each sub-panel so that we can have ten cases of α in each panel. More example data points could be found in Supplementary Figure S4, which has already been included in the Supplementary Materials submitted with our original manuscript.

By the way, our response to this comment also applies to Fig 2(d,e) that are drawn in similar way, although Fig. 2 was not specifically mentioned in your comment.

Comment#3

Could the authors comment on how their results compare with those in ref. 28, which observes a third—order LEP, but with non-Markovian dynamics?

Our Response:

We appreciate your valuable suggestion. We had also noticed the work in Ref. [28] investigating the EPs relevant to non-Markovian dynamics.

Both our work and Ref. [28] explore the high-order EPs. However, we investigate the third-order LEPs experimentally with Markovian dynamics but Ref. [28] studied theoretically for non-Markovian dynamics. If we would like to experimentally achieve a LEP with non-Markovian dynamics, we have to introduce strong dissipation in our system. Unfortunately, our effective two-level system, originating from a three-level system (see Figure 1d), could not reach a strong decay since the strength of 854-nm laser is required to be much smaller than the decay rate of the P state, see Appendix B of Ref. [50]. As such, it is hard to carry out the scheme considered in Ref. [28] in our experimental setup. We have mentioned this point in the last paragraph of the revised main text.

Comment#4

Fig. 3 a and b are not understandable in their present form. Please also mark the EPs where needed.

Our Response:

We apologize for this misunderstanding.

We have re-plotted Fig. 3(a, b) with thicker lines and short arrows to highlight the trajectories of the third-order LEP's movement. Some relevant sentences in the caption are also rewritten.

Moreover, since Fig 2 is plotted in a similar fashion, we have also updated it.

Comment#5

Hybrid-LEPs have a different meaning in recent literature (ref. 33) than what the the authors mean. The authors mean a hybrid between decay and Dephasing LEPs, while the well-known Ref. 33 introduces hybrid EPs between Hamiltonian and Liouvillian EPs. It is also strange to call Eq. (1) a hybrid Lindblad equation or (2) a hybrid Liouvillian because (1) is an ordinary Lindblad equation, just with two distinct processes.

Our Response:

We appreciate your comment very much since it reminds us of the possible misunderstanding of the phrase “hybrid-LEP” employed in our manuscript.

To avoid any confusion or misunderstanding, we have replaced “hybrid-LEP” by “Lindblad-LEP” throughout the revised manuscript, since our studied LEP, considering both decay and dephasing effects, originates from the standard Lindblad master equation.